# Myopia Alters the Structural Organization of the Retinal Vasculature, GFAP-Positive Glia, and Ganglion Cell Layer Thickness

**DOI:** 10.3390/ijms23116202

**Published:** 2022-05-31

**Authors:** Carol Lin, Abduqodir Toychiev, Reynolds Ablordeppey, Nefeli Slavi, Miduturu Srinivas, Alexandra Benavente-Perez

**Affiliations:** Department of Biological Sciences, SUNY College of Optometry, New York, NY 10036, USA; clin@sunyopt.edu (C.L.); atoychiev@sunyopt.edu (A.T.); rablordeppey@sunyopt.edu (R.A.); ns3377@columbia.edu (N.S.); msrinivas@sunyopt.edu (M.S.)

**Keywords:** myopia, astrocyte, vasculature, marmoset, neurovascular unit

## Abstract

To describe the effect of myopic eye growth on the structure and distribution of astrocytes, vasculature, and retinal nerve fiber layer thickness, which are critical for inner retinal tissue homeostasis and survival. Astrocyte and capillary distribution, retinal nerve fiber (RNFL), and ganglion cell layer (GCL) thicknesses were assessed using immunochemistry and spectral domain optical coherence tomography on eleven retinas of juvenile common marmosets (*Callithrix Jacchus*), six of which were induced with lens-induced myopia (refraction, Rx: −7.01 ± 1.8D). Five untreated age-matched juvenile marmoset retinas were used as controls (Rx: −0.74 ± 0.4D). Untreated marmoset eyes grew normally, their RNFL thickened and their astrocyte numbers were associated with RNFL thickness. Marmosets with induced myopia did not show this trend and, on the contrary, had reduced astrocyte numbers, increased GFAP-immunopositive staining, thinner RNFL, lower peripheral capillary branching, and increased numbers of string vessels. The myopic changes in retinal astrocytes, vasculature, and retinal nerve fiber layer thickness suggest a reorganization of the astrocyte and vascular templates during myopia development and progression. Whether these adaptations are beneficial or harmful to the retina remains to be investigated.

## 1. Introduction

Myopia is a significant risk factor for glaucoma, maculopathy, and choroidal neovascularization, among others [1,2]. All myopes, regardless of degree, are at an increased risk of visual impairment [1,2,3,4]. Despite the predicted global increase in myopia prevalence and its potential public health crisis in vision care [2,5], the mechanisms that lead to myopic degeneration and associated conditions remain unknown. There is a lack of early diagnostic markers for myopic pathology and no means to prevent disease progression [5,6].

The development and maintenance of a healthy retina relies on the neurovascular interplay between neuronal, vascular, and glial cells [7], which provide structural and nutritional support [7,8], regulate metabolism [7,9,10], neuronal debris phagocytosis [7,9,10], and ion and neurotransmitter homeostasis [10]. The neurovascular unit exerts a biphasic influence of degenerative and subsequent reactionary regeneration in systemic pathology that can be harmful at the acute phase and beneficial at the chronic phase [11]. During normal development and aberrant disease process, blood vessels, retinal astrocytes, Müller cells, microglia, and ganglion cells are in a reciprocal feedback loop [12]. For example, astrocyte numbers and distribution are determined by retinal capillary density and the amount of RNFL damage due to pathology [13,14]. Astrocyte reactivity and degeneration precedes ganglion cell degeneration and pathological neovascularization, respectively [15,16]. Müller cells, similar to astrocytes, respond to elevated intraocular pressure [17] and retinal injury [18], with GFAP upregulation that appears greater in areas closer to ganglion cells where astrocytes and Müller cell endfeet are located. Reactive Müller cells can affect neuronal activity due to their roles in synaptic and extracellular space regulation [19].

Myopic eyes experience blur because they are larger in size, which appear to result in a compromised vascular support to the inner retina. Lower central retinal artery blood velocities [20,21], a narrowing of the retinal vessel diameter [21], decreased capillary density [5], and larger foveal avascular zones [22] have been described in human myopic eyes with no degeneration. Due to the interrelationship between the neurovascular elements, these vascular changes might in turn compromise vascular and neuronal function, premeditate astrocytic reorganization, and be part of the etiological cascade of events that increases the risk to develop posterior pole complications [23,24]. However, the longitudinal effect of myopia on capillaries, astrocytes, and ganglion cells, and how they relate to each other, remains unknown.

Research with non-human primates (NHP) is a critically important link between animal studies and human treatments [25,26,27]. The common marmoset (*Callithrix jacchus*) has been established as an excellent NHP model for vision and neuroscience research because of its small size, fast development, ease in breeding and handling, and high optical quality eye with a diurnal foveated retina [28,29,30]. Common marmosets have successfully been induced with moderate and high myopia using negative contact lenses, following our well-established lens-induced myopia paradigm [28,29,30,31,32]. The strengths of our marmoset myopia model are the nonexistence of another NHP model of myopia that uses contact lenses to induce myopia, the controlled experimental conditions, and the ocular anatomy and physiology being directly comparable to human eyes. In this study, the structure and distribution of three key elements of the retinal neurovascular unit—astrocytes, co-localized superficial capillaries, and RNFL/GCL thickness—were studied in a NHP model of myopia to assess the neurovascular changes that eyes experience during myopia development and progression and how they relate to each other, with the ultimate goal to understand the effect of progressing myopia on the ocular tissues.

## 2. Results

### 2.1. Qualitative Characterization of Superficial Capillaries and Associated Astrocytes in Untreated Marmoset Retinas

Six myopic marmoset eyes were studied (age 200.3 ± 14.2 days). Five untreated age-matched juvenile marmosets were used as controls (age 232.2 ± 32.9 days). In control eyes, blood vessels were similar in width and shape within the fovea and parafovea (Figure 1B). The marmoset retina exhibited four vascular plexi in the foveal and parafoveal retina (Figure 1A, area 1 and 2): the radial peripapillary capillary plexus (RPC), superficial capillary plexus (SUP), intermediate capillary plexus (INT), and deep capillary plexus. The peripapillary retina had three vascular plexi (Figure 1A, area 3), while the peripheral retina had two vascular plexi (Figure 1A, area 4). The four vascular plexi had varying vessel diameters, (Figure 1B), with the vasculature becoming more regular in pattern with deeper plexi. The co-localization graphs in Figure 1A demonstrate the presence of different and distinct layers of vessels visible in the marmoset retina, in the retinal regions of this study. Four layers or vasculature are present in the parafoveal and foveal regions, three layers in the peripapillary region, and two layers in the peripheral region. There are two layers of corresponding astrocytes in the foveal and parafoveal regions, and one layer of astrocytes in the peripapillary and peripheral regions.

There were two layers of astrocytes in the foveal and parafoveal marmoset retina (Figure 1C), corresponding to the RPC and superficial vascular plexi. The astrocytes of the RPC were more numerous than those in the superficial layer (Figure 1C). In the peripapillary and peripheral marmoset retina, there was only one layer of astrocytes (Figure 1D,E). Representative astrocytes from control eyes can be seen in Figure 2B within panels 1, 3, and 5. These findings in control marmosets describe the physiological variations in vessel and astrocyte anatomy that were expected to be found in the optic nerve head, fovea, and periphery of untreated marmoset retinas.

The vasculature in focal areas closer to the optic nerve head contained vessel diameters of varying widths (Figure 2A, panel 1–3). The further away from the optic nerve head, the smaller and more uniform the widths of the retinal blood vessels are (Figure 2A, panels 4–6). Specifically, the average vein vessel width in focal regions 1–3 was 18.06 ± 2.9 microns, while the average artery width in the same regions was 15.74 ± 3.1 microns (*p* > 0.05). The average vein vessel width in focal regions 4–6 was 12.25 ± 3.4 microns, while the average artery width in the same regions was 9.95 ± 2.2 microns (*p* > 0.05). Images seen in Figure 2 are of the superficial vascular plexus; quantification was performed on both the RPC and superficial plexus.

String vessels are thin, non-functional connective tissue strands conserved across species that are remnants of capillaries. Vascular conditions such as diabetes and ischemia exhibit string vessels across various capillary beds of the body [33], and their numbers increase in normal aging but increase drastically in the presence and progression of vascular dysfunction. String vessels were identified in the superficial capillary plexus of marmoset retinas, and shown in Figure 3A as white arrows.

### 2.2. Superficial Capillaries Branch Less in the Retinal Peripapillary and Periphery of Myopic Marmosets

Representative images of the superficial vasculature are shown in Figure 3A. Myopic eyes had significantly less branching in the periphery and peripapillary region, and increased branching in the foveal retina, compared to controls (all *p* < 0.01; Figure 3B,C). Due to the magnification at which the branching analysis was performed, we were unable to distinguish arteries from veins. There were also a greater number of string vessels in the myopic parafoveal superficial capillary plexus than in controls (*p* < 0.01; Figure 3D). The retinal vasculature coverage, as measured by IB4 staining, was quantified and no difference were found between control and myopic marmoset retinas. The percentage area covered by capillaries in control and myopic animals decreased from the optic nerve head to the periphery. None of the regions differed between control and myopic marmosets (Table 1).

### 2.3. Myopic Eyes Have Reduced Astrocyte Cell Counts and Increased GFAP-Immunopositive Space

The number of astrocyte nuclei was lower in myopic than in control eyes in the superior, inferior, nasal, and temporal peripapillary and peripheral regions (Figure 4A,B,D, all *p* < 0.05). Despite the reduced astrocyte cell counts, the spatial coverage of astrocyte processes quantified as percentage GFAP-positive (GFAP+) immunostaining was significantly greater in the peripapillary but not the peripheral retina of myopes when compared to controls (Figure 4C, Peripapillary *p* < 0.05; Periphery *p* ≥ 0.70). There was no significant difference in the temporal retina percent GFAP+ staining (Figure 4E, Peripapillary *p* = 0.92, Periphery *p* = 0.38). The astrocytes in the parafoveal region appeared different depending on the layer which they are found (Figure 5A,B). The number of astrocytes in the foveal and parafoveal retina was significantly decreased in the myopic RPC vascular layer (Figure 5C, Fovea *p* < 0.001, Parafovea *p* < 0.001) and the myopic superficial vascular layer (Figure 5E, Fovea *p* < 0.001, Parafovea *p* < 0.001). The percentage GFAP+ immunostaining was greater in the RPC and superficial vascular plexi of myopic foveas and parafoveas (RPC Figure 5D, Fovea *p* < 0.05, Parafovea *p* < 0.05; superficial Figure 5E, Fovea *p* < 0.05, Parafovea *p* < 0.05). The increased GFAP-immunopositive staining appears to affect both astrocytes and Müller glia. While performing our 3D reconstruction of myopic retinal flatmounts, the GFAP staining did not appear uniform in areas corresponding to Müller cells. We cannot make a definitive statement on Müller cells as we did not counterstain with Müller cell markers in order to clearly distinguish astrocytes versus Müller cell GFAP staining.

The extent to which myopic retinal stretch and magnification affected astrocyte cell counts was corrected using a tangential equation. The retinal area for the average myopic retina after correcting for magnification was found to be similar to control eyes (0.64 mm in control eyes, 0.65 mm in myopic eyes), confirming that the effect of image magnification on the cell count calculations was minimal.

### 2.4. Untreated—But Not Myopic Eyes—Exhibit Thicker RNFL, Less GFAP Relative Frequency Index, and More Astrocyte Numbers

To confirm the validity of the SD-OCT scans to measure marmoset retinal thickness, an image of marmoset retinal histology (Figure 6A) is compared to a marmoset SD-OCT scan prior and post automated segmentation (Figure 6B,C respectively) (Iowa Reference Algorithms v3.6). A side-by-side magnified comparison of the marmoset retinal layers using retinal histology and SD-OCT scans can be seen in Figure 6D.

The averaged values of the parafoveal retinal nerve fiber thicknesses (RNFL) in control and myopic marmosets are depicted over representative SD-OCT marmoset fundus photographs (Figure 7A, left column). Segmentation calculations are shown in Figure 7A (right column). Representative images of the GFAP relative frequency index (RFI) can be seen in Figure 7B, showing the increase in GFAP+ staining through the thickness of the myopic retina compared to that of the control. The increased GFAP+ staining is not only limited to astrocyte-associated areas but also occurring in deeper layers, suggesting Müller cell involvement. The myopic RNFL was significantly thinner in the parafovea when compared to controls (Figure 7C: *p* < 0.05). The RFI of GFAP was significantly increased in the myopic peripheral superior, inferior, and nasal retinas (Figure 7D: Peripapillary *p* = 0.12, Periphery *p* < 0.05). The GFAP RFI was significantly increased in the myopic peripheral temporal retina (Figure 7E: Peripapillary *p* = 0.06, Periphery *p* < 0.05). GFAP RFI was significantly increased in the myopic parafoveal and foveal retinas (Figure 7F: Parafovea *p* < 0.001, Fovea *p* < 0.05).

## 3. Discussion

This study provides evidence of significant changes in three main retinal neurovascular elements and how they relate to each other in a NHP model of lens-induced myopia. When compared to age-matched controls, myopic marmosets had lower astrocyte counts in all retinal quadrants, increased GFAP-immunopositive staining, decreased capillary branching in the periphery, increased numbers of string vessels, and thinner RNFL. In this study, the inner retina’s capillary bed and co-localized astrocytes of marmosets were successfully identified and quantified in the marmoset, a NHP model that has captured the attention of neuroscientists due to its similarity to the human eye. The confocal images obtained from the marmoset retina show a four-layered capillary plexus (radial peripapillary, superficial, intermediate, and deep capillary plexi) and the presence of co-localized astrocytes, similar to human retinas [29,34,35].

### 3.1. Vascular Characterization

The marmoset superficial vascular layer includes the main arteries and veins that branch into smaller capillary vessels and form a narrowly stratified capillary network. The layout and branching pattern identified are comparable to human retinas, indicating a close evolutionary link and confirming marmosets as a good model to study retinal vasculature structure and function [34]. Marmosets with lens-induced myopia had lower blood vessel branching in the periphery but higher branching in the fovea when compared to controls. The myopic decline in peripheral branching suggests a reorganization of the vascular bed as myopic eyes grow and stretch. Reduced vessel branching has been associated with a decreased retinal blood supply in mouse eyes [36,37]. In our study, due to the magnification at which the branching analysis was performed, we were unable to distinguish arteries from veins. Thus, our discussion is limited regarding whether arteries or veins branched more in the myopic retinas. There is evidence that as myopic eyes elongate; they can become more prolate in shape with an asymmetric elongation along the horizontal axis [38], stretching the peripheral retina to a greater extent than the central retina. This growth pattern would explain the decreased peripheral but increased foveal branching found in this study. In order to maintain an adequate vascular supply to the increased myopic retinal area, capillary coverage and branching would have to expand. In this study, however, we identified a decrease in peripheral branching. The myopic decrease in peripheral vascular branching may suggest a compromised vascular and metabolic supply to the myopic periphery and potential state of hypoxia [39]. In fact, there is evidence in the literature that human myopic eyes exhibit non-perfused areas in the far retinal periphery [40]. This decrease in vessel branching might also indicate a form of capillary regression, which is known to lead to string vessel formation. String vessels are non-functional capillary strands caused by macrophages engulfing apoptotic endothelial cells [41]. The existence of these apoptotic endothelial cells has been associated with increasing age, decreased vascular endothelial growth factor (VEGF) concentration [42], hypoxia [43], and altered blood flow [44,45,46,47]. In this study, myopic eyes had a greater number of string vessels than control eyes in the parafoveal retina, suggesting that myopic retinal capillaries are experiencing capillary regression and string vessel formation [46,47,48,49]. While highly myopic eyes with thinner choroids have been described to have lower aqueous VEGF concentration [50], the relationship between string vessel formation, VEGF, and myopia progression remains unexplored.

### 3.2. Astrocyte Characterization

Studies on astrocyte distribution in the retinal plexi have been done in mice [51,52], mammals [51,53,54,55,56], cats [54], and humans [57,58,59], and describe mostly stellate-shaped astrocytes in the outermost retinal periphery and more elongated ones in the central retina [53,58,60,61]. Astrocytes in the primate retina are proportional in number to the thickness of the RNFL, and have the highest numbers at the optic nerve head [56]. This study findings suggest that astrocytes populate the marmoset retina, maintaining and establishing interactions with endothelial cells in the capillary walls, as previously described in other primates [34,55]. There were similar astrocyte numbers across the untreated marmoset retina, suggesting that astrocyte growth and development across the retina is fairly symmetrical, similarly to macaque, cat, and rabbit retinas [57,62,63]. Myopic eyes exhibited lower astrocyte numbers than controls in the peripapillary and periphery, and these numbers remained significant after correcting for magnification. Therefore, ocular magnification secondary to myopic stretch alone cannot explain the decrease in astrocytes identified in this study. This decrease in astrocyte counts suggests a reorganization and redistribution of the astrocyte template as a consequence of myopia development and sustained mechanical stress on astrocyte cell structure. Myopic growth is not symmetric and tends to occur to a greater extent in the periphery. Measurements of peripheral eye length would confirm whether peripheral scaling may be responsible for the lower numbers in the periphery. The lower astrocyte counts might also be related to the known association between astrocyte numbers and RNFL thickness (Lin CR, et al. IOVS 2021; 62.8: ARVO E-Abstract), which in this study and others was found to be thinner. There is evidence in the literature of astrocytes and Müller cells increasing GFAP expression during reactive gliosis and contributing to remodeling in various non-uniform ways [16]. In this study, a significant increase in GFAP-immunopositive staining was observed in the myopic parafovea, suggesting a mild astrocyte and Müller cell activation and a potentially compromised glial support to the ganglion cells of myopic eyes. Of particular importance is the role that ganglion cells, astrocytes and Müller cells play in mechanosensitivity [64], which is a feature of myopic stretch. Ganglion cell axonal damage is secondary to astrocyte, Müller cell, and microglia activation. They respond to mechanical stress, injury, and degeneration with structural and functional changes [65] that can be either detrimental or beneficial to axon survival and regeneration [66].

### 3.3. Astrocytes and the Vasculature

The existence and distribution of retinal astrocytes is closely related with the presence and distribution of retinal blood vessels [67]. Animal models of oxygen-induced retinopathy have described a reduced astrocytic network and highlighted the importance of astrocytes in the formation of retinal revascularization [36,68]. The decrease in astrocyte numbers observed in all areas of the myopic marmoset retina, in conjunction with the decrease in blood vessel branch points, supports the hypothesis that astrocytes might be affected by the vascular changes found. Whether the vascular changes may lead to hypoxia or decreased perfusion, and how this affects astrocytes, remains to be assessed [36]. This study did not evaluate retinal oxygenation, but it has been assessed by others; retinal arteriole and arterio-venous oxygen saturation is significantly decreased in highly myopic eyes with and without myopic retinopathy when compared to controls [69]. If this was confirmed, astrocytes may be responding to changes in oxygen demand and might be involved in the mechanisms leading to retinal myopic complications.

### 3.4. RNFL

Ocular pathologies exhibiting GCL and RNFL thinning with disease progression include glaucoma [70], macular degeneration [71], optic neuritis [72], and Alzheimer’s disease [73]. In this study, the RNFL was significantly thinner in myopic marmoset eyes and remained significant after correcting for the effects of magnification secondary to myopic stretch, suggesting that myopia affects ganglion cell axon distribution, which has been described by others in human eyes [71,74,75,76,77].

## 4. Methods

### 4.1. Marmoset Model of Myopia

Eleven juvenile marmoset eyes were studied, six of which were induced with myopia by imposing hyperopic defocus using full field negative single-vision soft contact lenses (−5D), and five were untreated controls. Earlier studies and statistical power analysis of the principle methods used indicated that 4 animals per experimental group provide an 80% power for our statistical analysis (*n* = 4 treated, *n* = 4 controls). All animal care, treatment, and experimental protocols were approved by the SUNY College of Optometry Institutional Animal Care and Use Committee (IACUC), the ARVO statement for the use of animals in ophthalmic and vision research, the US National Research Council’s Guide for the Care and Use of Laboratory Animals, the US Public Health Service’s Policy on Humane Care and Use of Laboratory Animals, and the Guide for the Care and Use of Laboratory animals. Identification, age, axial length, and refractive error of control and myopic marmosets are listed in Table 2. Untreated marmosets on average emmetropize to low degrees of myopia (−1.50 D) [31].

Cycloplegic refractive error (Rx, Nidek ARK-700A autorefractor, Nidek Co., LTD, Aichi, Japan), ocular axial length (AL, Panametrics, NDT Ltd., Waltham, MA, USA), and spectral domain optical coherence tomography (SD-OCT, Bioptigen SD-OCT; 12 × 12 mm^2^, 700 A-scans/B-scan x 70 B-Scans × 5 Frames/B-scan) were performed at baseline and at the end of treatment prior to enucleation. RNFL and GCL thickness were measured using the SD-OCT under anesthesia (alphaxalone, 15 mg/kg, IM), and were segmented and quantified using The Iowa Reference Algorithms (Version 3.6, Iowa Institute for Biomedical Imaging, Iowa City, IA, USA).

### 4.2. Enucleation, Dissection, and Flat-Mount Preparation

At the end of treatment, eyes were enucleated and placed in phosphate-buffered saline (PBS) (ThermoFisher, Waltham, MA, USA). Dissected retinas were fixed in Para-Formaldehyde (PFA) 4% in PBS (Santa Cruz Biotechnology, Dallas, TX, USA) for 40 min, washed five times for 30 min each with PBS, and incubated with 5% normal goat serum (ThermoFisher) and 0.5% TritonX (Sigma Aldrich, St. Louis, MO, USA) blocking buffer to avoid non-specific antibody binding. Following blocking, the retina was incubated with primary antibodies diluted in blocking buffer at 4 °C for 3 days. The primary antibodies used in this study were isolectin–Alexa 488 (1:100) (ThermoFisher), mouse anti-GFAP (1:500) (Sigma Aldrich), and rabbit anti-Sox 9 (1:1000) (Sigma Aldrich). After the incubation period, the retinas were washed five times for 30 min each with PBS and incubated with goat-anti mouse secondary antibody conjugated with Texas Red (1:500) (ThermoFisher) and goat-anti rabbit 647 (1:500) (ThermoFisher). The SuperFrost slides (ThermoFisher) were cleaned with ethanol. Retinas were inspected for any signs of debris, and consistent tissue thickness achieved by pinching and cutting vitreal remains. Retinas were plated and cover slips were placed on objectives with DAPI mounting medium (Vector Laboratories, Newark, CA, USA), were permitted to self-seal, and stored at −20 °C.

### 4.3. Confocal Microscopy and Image Acquisition

The immunochemical samples were imaged using the Olympus FV1200 MPE confocal microscope. The images were gathered, and the analyses were performed in a randomized manner by one blind investigator. Sixteen images (640 μm × 640 μm along the horizontal plane, and 10 μm along the vertical plane) were taken from each of the eleven retinas imaged. Multi-plane z-series were collected using a 20× objective, with each section spaced 1 μm apart. These 10 sections were processed by the confocal microscope to form a single z-stack of images subtending the whole specimen. Images were processed using Fiji software.

The distribution of astrocytes and co-localized blood vessels were assessed by imaging all four retinal quadrants (temporal, nasal, superior, and inferior) in the periphery, peripapillary, and parafoveal retina. This regional analysis was performed with the goal to identify local changes that might occur in myopic eyes due to their assymetric eye growth pattern. A representation of the locations evaluated can be seen in Figure 8A, which is a composite of 60 individual frames of a control marmoset retina (ID: C16, Right) taken at 4× magnification visualized with isolectin–488 using the Olympus confocal microscope. To ensure a detailed analysis, images were acquired at higher magnification in focal area 3 for the peripapillary region, and focal area 6 for the peripheral region. An independent foveal analysis was performed on the foveal and parafovea regions (boxes 1 and 2, Figure 8B). Detailed images of the foveal (Figure 8C) and parafoveal (Figure 8D) regions show the difference in anatomy between the two areas. Images were also acquired at focal areas 1–6 starting from the optic nerve head, as shown in Figure 8A, which represent focal distances away from the optic nerve head.

### 4.4. Image Analysis

Blood vessel analysis: Blood vessel branching points, as well as the presence and number of string vessels per image frame were manually counted for each frame on the branches of all orders and converted to number of branch points/mm^2^ and number of string vessels/mm^2^, respectively. The branch points and string vessels were quantified in the superficial capillary plexus. The percentage of retinal area covered by blood vessels was quantified using Fiji [78]. The image was split into its color channels to identify the green channel corresponding to IB4 (isolectin). The green channel image was made “binary”, (Process, Binary, Make Binary), then converted to mask (Process, Binary, Convert to Mask). The resultant image is white with black particles, and the black particles were summarized (Analyze, Analyze Particles, Ok). Fiji then gave objectively the percentage area covered by blood vessels.

Astrocyte quantification: The number of astrocyte nuclei was counted in every image frame using the Fiji cell counter function and converted to astrocytes/mm^2^. The image was split into its color channels to identify the red channel corresponding to GFAP. The red channel image was made “binary”, (Process, Binary, Make Binary), then converted to mask (Process, Binary, Convert to Mask). The resultant image is white with black particles, and the black particles were summarized (analyze, Analyze Particles, Ok). Fiji then gave objectively the percentage area of GFAP coverage, which we interpreted as GFAP-immunopositive staining and which we used to quantify astrocyte spatial coverage.

The increased GFAP+ immunostaining in the myopic retinas in our flatmount images probed us to perform three-dimensional reconstructions of the retinal tissue to see how GFAP distributes through the marmoset retina’s depth. The “3D stack function” on Fiji constructed a three-dimensional image from the flatmounts by flipping the z-stacks 90° on the X axis. We drew a vertical line horizontally across these 3D reconstruction images, of the same length and in the same area for all eyes. The “plot profile” function in Fiji then created a two-dimensional graph of pixel intensities along the line, with the x-axis representing distance in micrometers and the y-axis representing gray scale intensity. The “relative frequency index” (RFI was then calculated by averaging the values of minimum GFAP intensity subtracted from the average maximum GFAP intensity in all areas. This value was gathered in the myopic and control peripapillary retina, where the most myopic GFAP-immunopositive staining was found.

### 4.5. Correction for Magnification

The extent to which myopic retinal stretch affected the blood vessel and astrocyte calculations was determined using a tangential equation. For the OCT calculations, the diameter of the ETDRS grid circle was adjusted for ocular magnification using the formula: 𝑅_N_ = (𝑅 × 𝐴𝐿)/9.62, where *R* is the radius of the ETDRS grid and *AL* is the axial length of the marmoset. The value 9.62 is the average axial length in millimeters of adult control marmosets in our lab.

### 4.6. Statistical Analysis

Data were assessed for normality and analyzed using an analysis of variance (ANOVA) at the level of α = 0.05 using OriginPro 2021b software (OriginLab, Northampton, MA, USA).

## 5. Conclusions

The results from this study suggest that myopic eye growth affects the architectural template of three key retinal neurovascular elements: blood vessels, astrocytes, and ganglion cells. The restructuring and reorganization observed in NHP myopic eyes may reflect a dynamic adaptation to the changing environment triggered by myopic eye growth and stretch, and may be part of a beneficial adaptive chronic response to support neurons during ocular growth [11]. Alternatively, it could represent a detrimental response indicating the beginning of a compromised structural vascular support to the inner retina, affecting vascular and astrocyte function and leading to an alteration in the ability to regulate local ions, neurotransmitters, and metabolites, as well as affecting neural function [11]. The changes in astrocyte numbers found in the myopic retina suggest that myopic retinal stretch triggers an adaptive response on astrocytes, which might include a lengthening of their cell bodies and axons in an effort to provide adequate structural and functional support to the vasculature and RGCs. This type of response has been described before in astrocytes exposed to artificially induced mechanical tension, which formed living scaffolds to guide neuroregeneration [79]. However, since astrocytes and retinal capillaries are crucial to maintain RGC axonal viability [26,80], the reduced astrocyte numbers observed in myopic eyes, along with the increased GFAP-immunopositive staining, reduced capillary branching, and increased numbers of string vessels, might be affecting neuroretinal function. In particular, the increased GFAP-immunopositive staining identified in the peripapillary and parafoveal myopic retina, along with the RNFL thinning observed in myopic eyes, might be a sign of early glial reactivity. The space occupied by neural axons lost due to injury or degeneration is generally filled by a glial scar, and predominantly involves hypertrophic astrocyte processes [51]. Astrocytes can become reactive to mechanical and chemical stimuli such as elevated intraocular pressure, increased mechanical tension [79], and excitotoxicity [16]. These changes reflect the biphasic nature attributed to astrocytes [11], which includes degenerative and regenerative properties at different phases after retinal stress or injury.

In conclusion, this study confirms the feasibility of the marmoset as an experimental model to study the retinal neurovascular unit. The aim was to evaluate in detail the effect of myopic eye growth on the structure and distribution of three key neurovascular elements: blood vessels, astrocytes, and ganglion cell/retinal nerve fiber layer thickness. This study confirms that myopic eyes without pathology exhibit changes in all three neurovascular elements, suggesting that the neurovascular unit may be affected by myopic mechanical stretch and elongation. Whether this neurovascular adaptation is beneficial or harmful, and whether its function is altered in disease, remains to be investigated. Future studies intend to evaluate quantitatively the morphology of astrocytes and how astrocyte phenotype may relate to retinal disease.

## Figures and Tables

**Figure 1 ijms-23-06202-f001:**
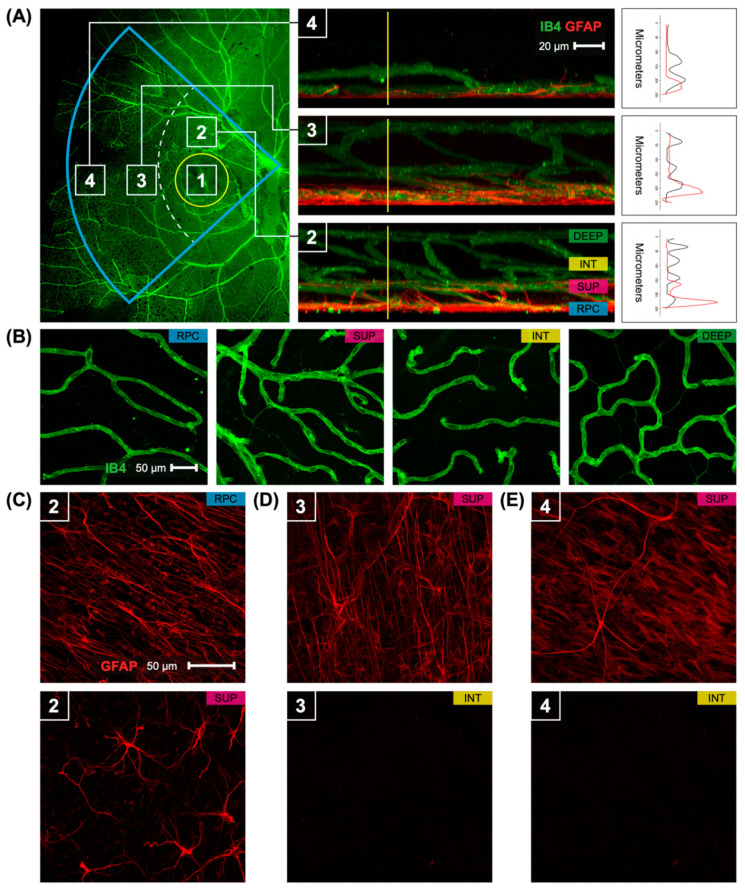
Retinal vasculature and astrocytes distribution in the temporal part of the control marmoset. (**A**) An image of the temporal part of the retina (**left**) is visualized with isolectin (green) and highlighted in a blue insert. Yellow circle indicates the location of the foveal region. Numbers in the white boxes represent areas (1: foveal, 2: parafoveal, 3: peripapillary, 4: periphery) that were used for 3D reconstructions. Reconstructed images (**middle**) from areas 2, 3, and 4 show the distribution of the inner vasculature (green) and astrocytes (red). The four vascular plexi are the radial peripapillary capillary (RPC), superficial (SUP), intermediate (INT), and deep (DEEP) plexi. Scale, 20 µm. A vertical line (yellow) is an intensity profile (**right**) used to show the colocalization of vasculature (black) with astrocytes (red). (**B**) Representative images of the retinal vasculature (green) acquired from the parafoveal area show all vascular layers. Scale, 50 µm. (**C**–**E**) An image of the retinal astrocytes (red) in areas 2, 3, and 4. Scale, 50 µm. In area 2 (parafovea) astrocytes are distributed in two vascular layers RPC and SUP layers. In other areas (3, 4), astrocytes are found only in the superficial layer.

**Figure 2 ijms-23-06202-f002:**
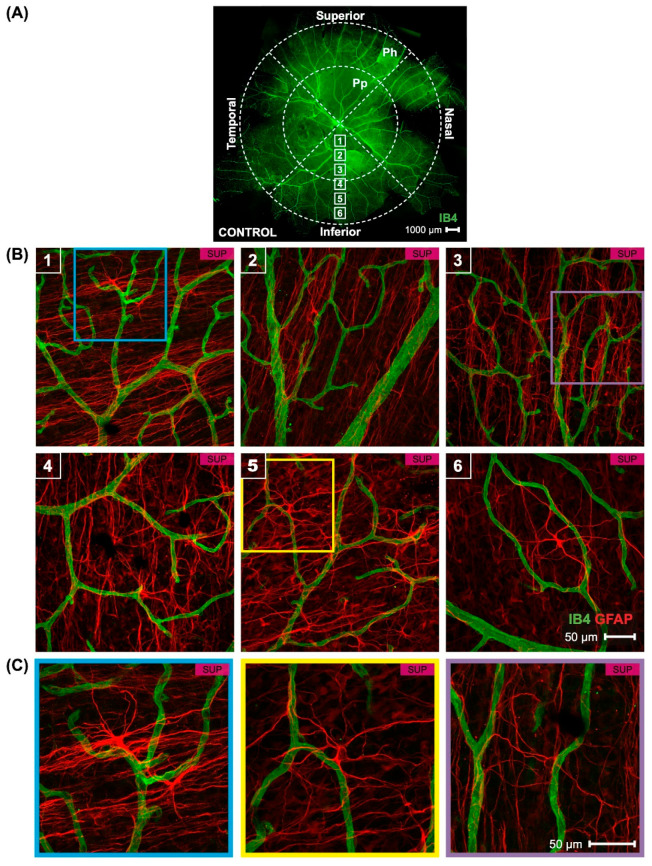
Visualization of the superficial astrocytes and vasculature in the control marmoset retina, in focal areas near the optic nerve head to the far periphery. (**A**) A complete map of the retinal vasculature (green) in a control marmoset (ID: C16 Right) where the peripapillary (Pp) and peripheral regions (Ph) are indicated. The yellow box represents the foveal region. White boxes represent focal areas away from the optic disc to periphery. Superior, inferior, nasal, and temporal quadrants of the retina are shown. (**B**) Representative images of the marmoset retina taken from the superficial layer showing the vasculature (green) and astrocytes (red) in focal areas of the same control marmoset at 40× magnification. (**C**) Astrocytes from specific focal areas are shown at 60× magnification with colored boxes (blue in panel 1; pink in panel 3; yellow in panel 5). Each panel represents a typical finding from a sample of five control marmosets.

**Figure 3 ijms-23-06202-f003:**
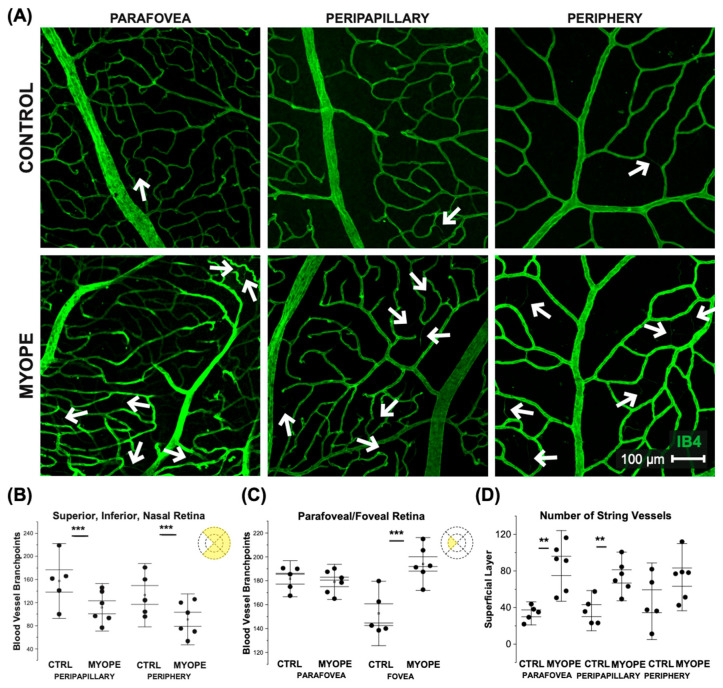
Vascular alterations of the myopic marmoset retina, shown in focal areas 2, 4, and 6 from the optic nerve head. (**A**) Representative images of superficial capillary structure in the parafoveal, peripapillary, and peripheral regions of a control marmoset (ID tag: C16 Right) and myopic marmoset (ID tag: O17), taken at 20× magnification. Vasculature is labeled with IB4 (green). White arrows point to string vessels found in the marmoset retinas. ** = *p* < 0.01, *** = *p* < 0.001 (**B**) Analysis of the blood vessel branch points in the peripapillary region and peripheral superficial vascular layer of the superior, inferior, and nasal retina. Data shown as a I-graph box plot for control (*n* = 5) and myopic (*n* = 6) marmoset retinas where the inner box lines represents standard error (SE) and outer lines/whiskers represents standard deviation (SD). The number of branch points were significantly different in the peripapillary and peripheral regions. Peripapillary *p* < 0.01, Peripheral *p* < 0.01. (**C**) Analysis of the capillary branch points in the fovea and parafovea region (control *n* = 5, myope *n* = 6). A significant increase in branching was seen in the myopic eye in the foveal regions, with no change noted in the parafoveal myopic retina. Fovea *p* < 0.01, Parafovea *p* = 0.16. (**D**) Analysis of the number of string vessels in the superficial capillary plexus (SCP). A significant increase in the number of string vessels in the myopic parafoveal SCP was noted (*p* < 0.01) and peripapillary SCP (*p* < 0.01). No significance was found in the peripheral region (Periphery *p* = 0.62).

**Figure 4 ijms-23-06202-f004:**
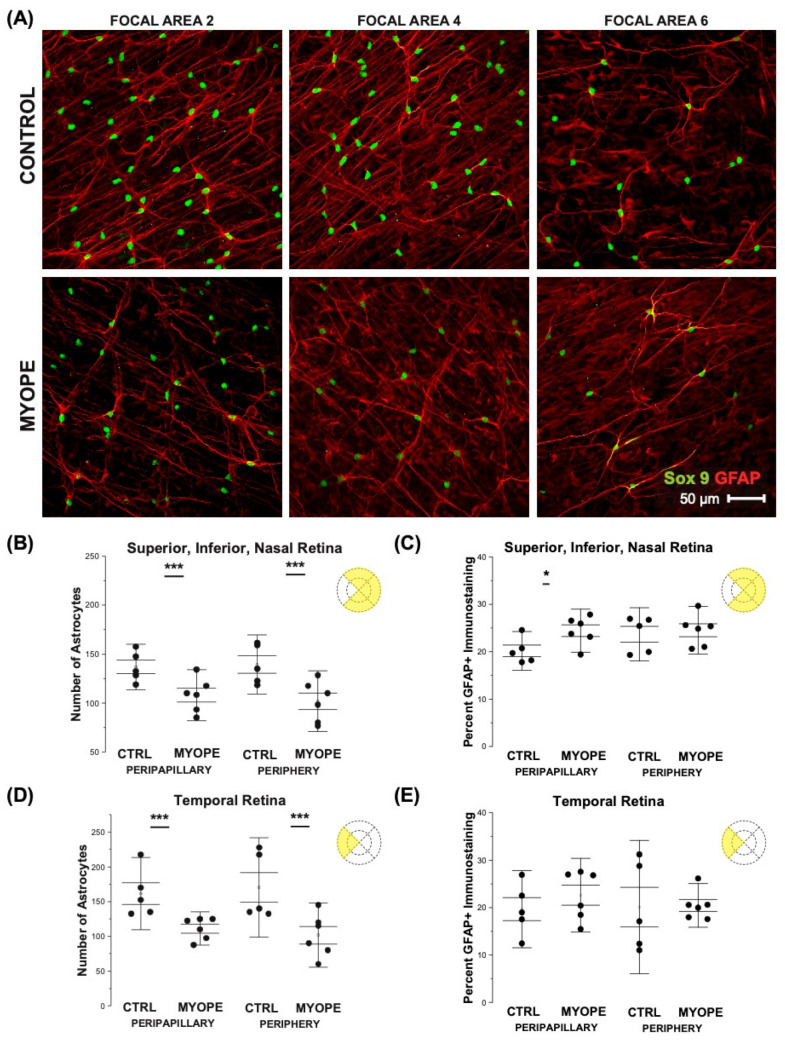
Myopic marmoset retinas show decreased numbers of astrocytes and increased GFAP-immunopositive staining in the peripapillary and peripheral retina, excluding the foveal region. * = *p* < 0.05, *** = *p* < 0.001. (**A**) Representative images of superficial astrocytes in the focal areas 2, 4, and 6 of control (ID tag: C16 Right) and myopic (ID tag P17 Right) marmosets. Images are taken at 40×, and astrocytic cell nuclei and bodies/processes in the retinas were labeled with Sox 9 (green) and GFAP (red) markers, respectively. (**B**) Shows analyses performed for number of astrocytes in the superior, inferior, and nasal retina. Data shown as a I-graph box plot for control (*n* = 5) and myopic (*n* = 6) marmoset retinas where the box represents SE and whiskers signify SD in (**B**) through (**E**). The number of astrocytes decreased significantly in the myopic retina in the superior, inferior, and nasal regions (Peripapillary *p* < 0.001, Peripheral *p* < 0.001) (**C**) The percentage of GFAP-immunopositive staining (GFAP+) increased significantly in the myopic peripapillary region of the superior, inferior, and nasal retina (Peripapillary *p* < 0.05, Peripheral *p* = 0.70). (**D**) The number of astrocytes in the temporal retina decreased significantly in myopic eyes (Peripapillary *p* < 0.001, Peripheral *p* < 0.001), similar to the other regions of the retina. (**E**) The percentage of GFAP+ immunostaining did not show any significant change between the temporal control and the myopic peripapillary region and periphery region (Peripapillary *p* = 0.92, Peripheral *p* = 0.38).

**Figure 5 ijms-23-06202-f005:**
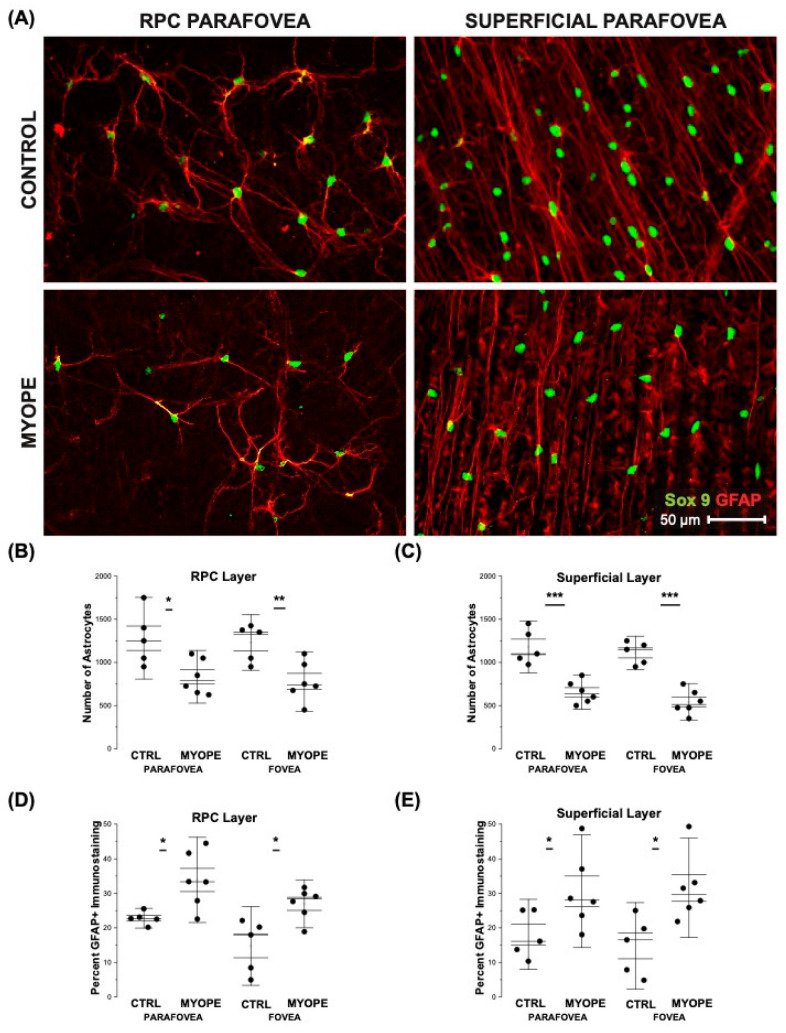
Myopic marmoset retinas showed decreased numbers of astrocytes and increased GFAP-immunopositive staining in both the foveal and parafoveal region. * = *p* < 0.05, ** = *p* < 0.01, *** = *p* < 0.001. (**A**) Representative images of RPC and superficial layer astrocytes in the parafoveal region of control and myopic marmosets (control ID tag: H16 Right) and myopic (ID tag: P17 Right). Images were taken at 40×, and astrocyte cell nuclei and body/processes in the retinas were labeled with Sox9 (green) and GFAP (red) markers, respectively. (**B**) Shows analysis performed for the number of astrocytes in the foveal and parafoveal regions (control *n* = 5, myopic *n* = 6). Data is shown as a box plot with SE as the box and SD for whiskers in (**B**–**E**). The number of astrocytes in the RPC vascular plexus of both the foveal and parafoveal regions decreased significantly in myopic eyes (Fovea *p* < 0.05, Parafovea *p* < 0.05). (**C**) A significant increase in the percentage of GFAP+ immunostaining was found in the RPC vascular plexus of same areas in the myopic retina (Fovea *p* < 0.05, Parafovea *p* < 0.05). (**D**) The number of astrocytes in the superficial vascular plexus of both the foveal and parafoveal regions decreased significantly in myopic eyes (Fovea *p* < 0.001, Parafovea *p* < 0.001). (**E**) The percentage of GFAP+ immunostaining was significantly increased in the superficial vascular plexus in myopic foveal and parafoveal retinas (Fovea *p* < 0.05, Parafovea *p* < 0.05).

**Figure 6 ijms-23-06202-f006:**
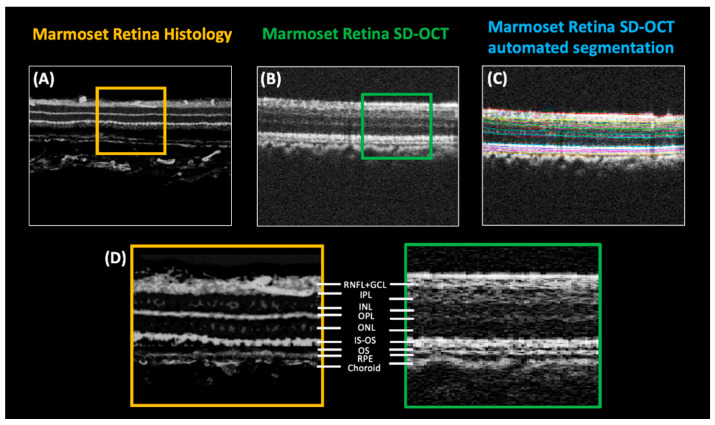
(**A**) A control marmoset retinal histology slice (ID tag H16 Right), compared to (**B**) a control marmoset SD-OCT scan (ID tag C16 Right). (**C**) An example of automated OCT segmentation from Iowa Reference Algorithms v3.6, performed on the same OCT scan from (**B**). (**D**) A side-by-side of histology and OCT scan, at higher magnification, of (**A**,**B**). Yellow (from histology) and green (from OCT) squares indicate areas that have been magnified to allow a direct comparison of the retinal layers.

**Figure 7 ijms-23-06202-f007:**
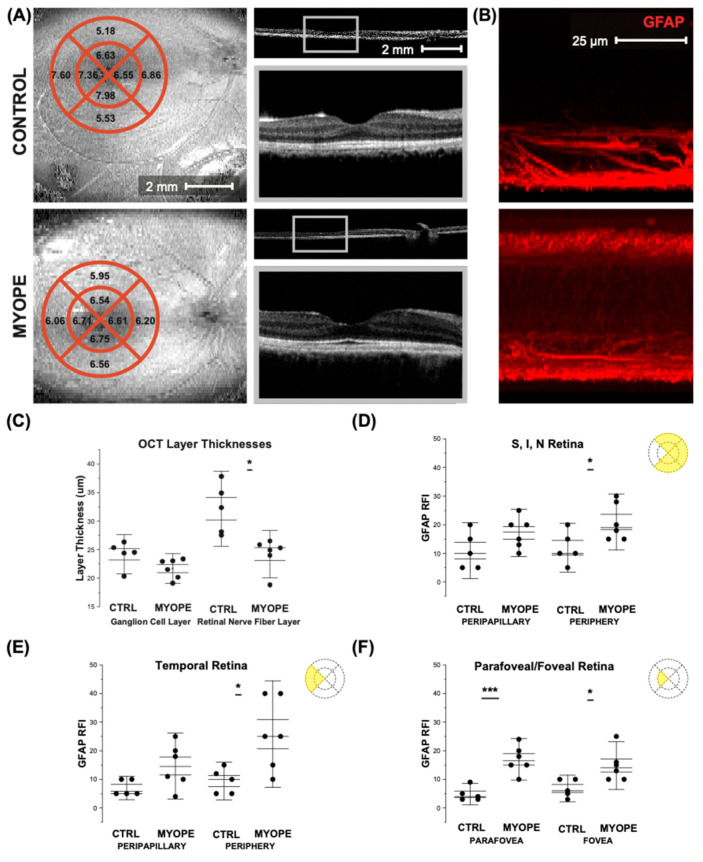
OCTs from the control and myopic marmoset. The graphs show differences in RNFL layer thickness, and the GFAP RFI of different retinal regions to be higher in the myopic retina. * = *p* < 0.05, *** = *p* < 0.001. (**A**) Average ganglion cell layer thicknesses in micrometers (mm) in the quadrants around the fovea of control (**top left panel**) and myope (**bottom left panel**) marmosets, as measured with SD-OCT. Representative image of en face optical coherence tomography around the fovea of a control marmoset (**top left**: ID tag C16 Right) and a myopic one (**bottom left**: ID tag O17 Left). Representative cross-sectional scan images of the fovea of the same marmosets can be seen for the control marmoset (**top right panel**, **top**) and myopic marmoset (**bottom right panel**, **top**), with close-ups of the cross-sectional scan images to be found in the top grey box for the control (**top right panel**, **bottom**) and the grey box for the myope (**bottom right panel**, **bottom**). (**B**) Representative GFAP RFI images of the superficial retina in a control (**top**: H16 Right; **bottom**: P17 Right), showing increased GFAP RFI in the myopic retina when compared to the control. (**C**) Analysis of the ganglion cell layer thickness showed that the myopic GCL was no different to the control GCL thickness in the parafoveal retina (*p* = 0.13). However, there is a significant decrease in the myopic RNFL thickness, compared to that of the control RNFL thickness (*p* = 0.04). (**D**) GFAP RFI was significantly increased in the myopic peripheral superior, inferior, and nasal retinas (Peripapillary *p* = 0.12, Periphery *p* < 0.05) (**E**) GFAP RFI was significantly increased in the myopic peripheral temporal retina (Peripapillary *p* = 0.06, Periphery *p* < 0.05). (**F**) GFAP RFI was significantly increased in the myopic parafoveal and foveal retinas (Parafovea *p* < 0.001, Fovea *p* < 0.05).

**Figure 8 ijms-23-06202-f008:**
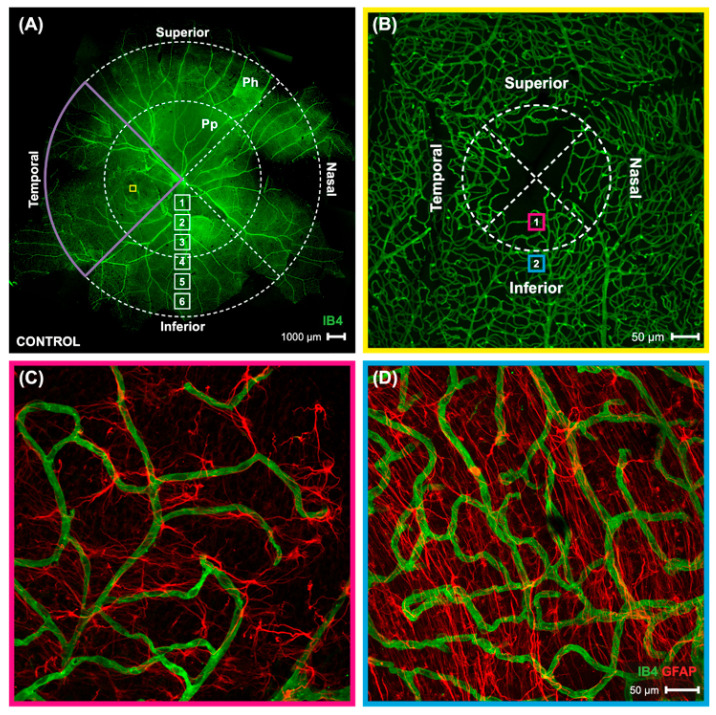
A map of the vasculature of the whole mount marmoset retina, and images of foveal and parafoveal region. (**A**) Shows a complete map of the retinal vasculature (green) obtained from a control marmoset (ID: C16 Left). The retinal vasculature was visualized with conjugated IB4-488. Images were acquired at 4× magnification and stitched in Photoshop. Location of peripapillary region (Pp) and peripheral region (Ph) described in this study is shown. Yellow box represents the foveal region, seen in panel 8B. White boxes represent focal areas away from the optic disc to periphery. Boxes labeled “Pp” represent locations where peripapillary location images were taken, while boxes labeled “Ph” represent locations where peripheral location images were taken. Superior, inferior, nasal, and temporal quadrants of the retina are shown. (**B**) A representative image of the vasculature (green) in the foveal region of the control marmoset (ID: H16 Right), taken at 10× magnification. Pink box labeled “1” represents the foveal region, while blue box labeled “2” represents the parafoveal location. (**C**,**D**) shows the anatomy of the astrocytes (red) and vasculature (green) in the fovea (8B, pink box) and parafovea (8B, blue box) region. Images were acquired at 40× magnification and astrocytes were visualized with GFAP marker.

**Table 1 ijms-23-06202-t001:** Percentage area covered by capillaries in controls and myopic marmosets (average ± SD).

	Control	Myope	*p*-Value
Superior Periphery	14.3 ± 3.7	14.0 ± 3.7	0.89
Superior Peripapillary	24.3 ± 15.2	18.5 ± 7.0	0.42
Inferior Periphery	15.2 ± 4.9	13.7 ± 5.0	0.63
Inferior Peripapillary	19.3 ± 7.0	15.4 ± 5.5	0.35
Temporal periphery	14.8 ± 5.3	13.4 ± 5.5	0.69
Temporal Peripapillary	17.8 ± 5.4	17.4 ± 4.4	0.90
Nasal Periphery	14.7 ± 3.9	11.6 ± 4.7	0.26
Nasal Peripapillary	23.1 ± 5.0	17.3 ± 6.8	0.06

**Table 2 ijms-23-06202-t002:** Treatment started at 10 weeks of age (72.0 ± 5.0 days) following our established protocol [30,31,32]. Lenses were inserted daily in the morning between 8–10 am when lights were turned on in the animal room (700 lux) and removed 9 h later at lights off each day (9 h light/15 h dark) [28,30]. Contact lenses had 6.5 mm diameter, 3.6/3.8 mm base curve, were made of methafilcon A (55% water content, DK: 17), fit 0.10 mm flatter than the flattest keratometry measurement, and were assessed using an ophthalmoscope. No corneal complications were observed in any of the animals treated in this or any earlier studies with marmosets [28,30].

Control ID, Eye	Age (Days)	Gender	Axial Length (mm)	Refractive Error (D)	Myope ID, Eye	Age (Days)	Gender	Axial Length (mm)	Refractive Error (D)
C16, Right	268	Female	10.259	−0.66	B17, Right	214	Female	10.900	−7.93
C16, Left	268	Female	10.241	−0.13	B17, Left	214	Female	10.894	−7.97
G16, Left	215	Male	10.279	−1.15	O17, Right	204	Male	10.492	−7.28
H16, Right	205	Female	10.286	−0.63	O17, Left	204	Male	10.212	−3.91
H16, Left	205	Female	10.307	−1.12	P17, Right	183	Female	10.554	−7.96
					P17, Left	183	Female	10.464	−3.08
AVG ± SD	232.2 ± 32.9		10.27 ± 0.03	−0.74 ± 0.4	AVG ± SD	200.3 ± 14.2		10.61 ± 0.3	−7.01 ± 1.8
	*p* > 0.05		*p* < 0.05	*p* < 0.01					

## Data Availability

The data that support the findings of this study are available from the corresponding author upon reasonable request.

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
