# Peer review of "Myopia Alters the Structural Organization of the Retinal Vasculature, GFAP-Positive Glia, and Ganglion Cell Layer Thickness"

_ijms, 2022, doi:10.3390/ijms23116202_

Round 1
Reviewer 1 Report
Summary: In the present article, the authors have used marmoset as a model to study the changes that occur in the retinal neurovascular architecture in lens-induced myopia. This is a significant topic as myopia is a huge risk factor for several other conditions including glaucoma, maculopathy, etc. They have found that myopic marmosets have reduced number of astrocytes in all the quadrants of the retina, decreased capillary branching in the periphery, increased numbers of string vessels, thinner RNFL but not associated between the reduction in astrocytes counts and RNFL thickness. Their conclusion is that these changes that happen due to myopia result from a reorganization of the astrocytes and vascular templates during myopia development and progression. The article is very well-presented, and all the sections look good to me. I only have a few comments.
- Why is there no quantification of the data in fig 2A?
- Can the authors show a low mag image in fig. 2 to show where the optic nerve head is and how different regions lie relative to that?
- Why are the authors using Sox 9 to normalize? Do the levels of Sox9 stay constant during myopia?
Author Response
We thank reviewer #1 for their comments.
In the present article, the authors have used marmoset as a model to study the changes that occur in the retinal neurovascular architecture in lens-induced myopia. This is a significant topic as myopia is a huge risk factor for several other conditions including glaucoma, maculopathy, etc. They have found that myopic marmosets have reduced number of astrocytes in all the quadrants of the retina, decreased capillary branching in the periphery, increased numbers of string vessels, thinner RNFL but not associated between the reduction in astrocytes counts and RNFL thickness. Their conclusion is that these changes that happen due to myopia result from a reorganization of the astrocytes and vascular templates during myopia development and progression. The article is very well-presented, and all the sections look good to me. I only have a few comments.
Why is there no quantification of the data in fig 2A?
Thank you for your comments. Figure 2A shows representative images of the superficial vasculature and astrocytes. Quantification is described in figures 3, 4 and 5.
Can the authors show a low mag image in fig. 2 to show where the optic nerve head is and how different regions lie relative to that?
We have included a low-magnification image within figure 2 (Figure 2A) to show where the optic nerve head and where the different focal regions are located in the retina.
Why are the authors using Sox 9 to normalize? Do the levels of Sox9 stay constant during myopia?
The levels of Sox9 decrease during myopia as shown in Figures 4 and 5. Therefore, we have removed Sox9 to normalize and presented independently astrocyte cell counts using Sox9, and astrocyte coverage using GFAP immunopositive staining. We have modified our methods, results, and discussion to indicate these changes.
Reviewer 2 Report
This manuscripts looks at the effects of myopia in the organization of retinal astrocytes and vasculature and ganglion layer thickness. Myopia was caused in common marmosets with lenses.
The aim of the study is of interest and it is nicely modeled in non-human primates. Imaging is of very good quality. However, my main concern with the study is that many of the descriptions presented in the results are qualitative observations. There are many statements with no quantification to support them. This is very clear in the first section of the results, I show here a few examples:
- Lines 80-81: "The four vascular plexi had varying vessel diameters, morphology, and distribution (Figure 1B), with vasculature becoming denser and more regular in shape with deeper plexi." This is stated without any quantification.
- Lines 102-103: "The vasculature in focal areas closer to the optic nerve head contained vessel diameters of varying widths". Again without any quantification.
- Lines 119 to 124: "Elongated astrocytes had smaller sized bodies than stellate astrocytes and were polygonal in shape. GFAP immunopositive staining per astrocyte was increased in the fovea, parafovea, and the peripapillary regions of the retina. The astrocytes in these regions were more dense, elongated and linear as they exited the optic nerve head (Figure 2A, panels 1-3), and became scarcer and more stellate as they approached the periphery (Figure 2A, panels 4-6)." Again, not a single quantification.
Another main concern is that, when the authors actually quantify different parameters, n numbers are very low, which gives no confidence in the significance of the results. n numbers are usually around 5, and are even of 3 in some case (see Figure 4E) with extreme error bars. No proper conclusions can be extracted from this data.
Of high concern is the fact that with 5 control animals and 6 myope animals in figures 5 or 6 the authors show graphs with n numbers of 18 or so... where is this data coming from?
Also, in the methods the authors indicate that power calculations suggest the use of an n = 8, but they go for groups of 6 or 5....
I am sorry, but the data is not reliable and the statements and conclusions are not supported by good quantitative data.
Also, the ages of the control and myopic animals are very different (a month aprox.). Could this affect the results? They do not have age matched controls for the experiments.
In figure 1, there are some colocalization graphs but these data is not explain or provided in the results.
I recommend rejection.
Author Response
We thank reviewer #2 for their comments.
This manuscript looks at the effects of myopia in the organization of retinal astrocytes and vasculature and ganglion layer thickness. Myopia was caused in common marmosets with lenses.
The aim of the study is of interest and it is nicely modeled in non-human primates. Imaging is of very good quality. However, my main concern with the study is that many of the descriptions presented in the results are qualitative observations. There are many statements with no quantification to support them. This is very clear in the first section of the results, I show here a few examples:
Lines 80-81: "The four vascular plexi had varying vessel diameters, morphology, and distribution (Figure 1B), with vasculature becoming denser and more regular in shape with deeper plexi." This is stated without any quantification. Lines 102-103: "The vasculature in focal areas closer to the optic nerve head contained vessel diameters of varying widths". Again without any quantification. Lines 119 to 124: "Elongated astrocytes had smaller sized bodies than stellate astrocytes and were polygonal in shape. GFAP immunopositive staining per astrocyte was increased in the fovea, parafovea, and the peripapillary regions of the retina. The astrocytes in these regions were more dense, elongated and linear as they exited the optic nerve head (Figure 2A, panels 1-3), and became scarcer and more stellate as they approached the periphery (Figure 2A, panels 4-6)." Again, not a single quantification.
We agree with the reviewer; however the aim of our work was to perform both a qualitative and quantitative analysis of superficial astrocyte and vessel morphology and distribution, and how this template is affected by myopia development and progression in relation to RNFL thinning. The first section of our results describes qualitatively the morphology of the superficial capillaries and astrocytes. Methods to quantify astrocyte morphology, especially for elongated astrocytes, in the parafoveal and peripapillary regions we have studied are being developed, but unfortunately these methods will take time. We have therefore modified the sections to make it clear that this is a qualitative study and added “qualitative” to the title of the section. Quantification for these figures are shown in figures 3-5 and 7.
Another main concern is that, when the authors actually quantify different parameters, n numbers are very low, which gives no confidence in the significance of the results. n numbers are usually around 5, and are even of 3 in some case (see Figure 4E) with extreme error bars. No proper conclusions can be extracted from this data. Of high concern is the fact that with 5 control animals and 6 myope animals in figures 5 or 6 the authors show graphs with n numbers of 18 or so... where is this data coming from?
Earlier studies and statistical power analysis of the principle methods used indicate that 4 animals per experimental group provide an 80% power for our statistical analysis (n=4 treated, n=4 controls). Additionally, our work incorporates a non-human primate (NHP) model. Other studies using NHP models also have small sample due to practical constraints: their care requires significant and unique infrastructure and resources, and requires enriched environments with highly trained staff (Huber, Hillary et al., Journal of developmental origins of health and disease, 11.3, 2020). We have successfully developed repeatable immunostaining protocols and assessed the within and between session repeatability of our cell count calculations; the 95% confidence interval (CI) falls within 7 cells. This variability is significantly smaller than the changes we are describing. To further decrease variability, we quantified each flatmount three times and calculated the average.
In addition, thank you for noticing the discrepancy in the sample size of figure 6 (now figure 7). We had originally plotted data for each retinal sub-region analyzed rather than the average per animal, therefore resulting in an n of 20 for controls (n =5 x 4 regions) and an n of 24 for myopes (n = 6 x 4 regions). We have now corrected this.
Line 706-709: [Earlier studies and statistical power analysis of the principle methods used indicate that 4 animals per experimental group provide an 80% power for our statistical analysis (n=4 treated, n=4 controls).]
Also, in the methods the authors indicate that power calculations suggest the use of an n = 8, but they go for groups of 6 or 5....
We apologize for the lack of clarity. By n=8, we meant 4 control marmosets and 4 treated marmosets.
Also, the ages of the control and myopic animals are very different (a month aprox.). Could this affect the results? They do not have age matched controls for the experiments.
As indicated in table 2, the average age for controls and treated animals was not different (controls: 232.2±32.9 days, myopes 200.3±14.2 days, p>0.05).
In figure 1, there are some colocalization graphs but these data is not explain or provided in the results.
Figure 1 visualizes the astrocyte and vascular templates in the marmoset retina. We observed four vascular plexi and the existence of two astrocyte layers in the parafoveal region in the marmoset retina, which had not been described to date. It is outside of our expertise to quantify the degree of colocalization and outside the scope of this work.
Line 100-107: [The co-localization graphs in Figure 1A demonstrate the presence of different and distinct layers of vessels visible in the marmoset retina, in the retinal regions of this study. Four layers or vasculature are present in the parafoveal and foveal regions, three layers in the peripapillary region, and two layers in the peripheral region. There are two layers of corresponding astrocytes in the foveal and parafoveal regions, and one layer of astrocytes in the peripapillary and peripheral regions.]
Reviewer 3 Report
Lin and co-authors investigated vasculature architecture and astrocyte behavior in a non-human primate model for myopia using majoritarily histological approaches. In their study, the authors found that the number of GFAP-positive cells are decreased in marmosets with myopia, and that this might be directly associated to a thinning of the RGL. They also found that vasculature branching points were reduced during myopia.
I personally find the study interesting and well-structured, and figures are organized and appealing. Understanding the neurovascular unit in disease is of relevance, and in the context of myopia still remains an understudied topic, especially using experimental primate models. The paper is well written and easy to follow with the sequence of results as displayed.
While the authors fall into a bit old-fashioned and fundamental approaches based majoritarily in histology (with the exception of Figure 6), I think that the data are of interest. However, most conclusions on astrocytic behavior were based on GFAP expression and this should be something to take cautiously in consideration.
Scientists are still struggling to characterize in-depth the nature and biology of astrocytes in the retina. While I agree with the conclusions of the authors based on the data shown, GFAP has been widely also associated to gliosis in Müller cells, another glial partner present in much higher numbers than astrocytes in the retina. Throughout the manuscript (results and interpretation), this fact was largely neglected or omitted. In Figure 6B, for instance, despite the much lower resolution of GFAP, it is very clear how GFAP staining is not limited to the defined astrocytic zones but also following a structure characteristic of Müller glia. Among their multiple functions, Müller glia act as scaffold in the retina, and are especially resilient cells to mechanic stress. A study relying only on GFAP associated to astrocytes and neglecting Müller glia is, from my perspective, an important bias that should be minimally (at least) addressed.
I have several specific comments aiming at improving the quality of the manuscript:
- Table 1 should display also the sex of the subjects. Sex in Medicine has been a largely ignored variable for centuries and it is in our hands to start amending the lack of knowledge. Consequently, authors should look at possible sex differences in their study.
- Table 1: Given that all ID tags are referenced in the table, the authors could omit part of the information (e.g. age, RX) when using the ID tag in the figure legends.
- I suggest that the authors indicate in results (when first mentioned Table 1) the threshold to consider myopia in the subjects (rx>XX)
- Figure 1B: The images are appealing but this should be complemented with semi-automatic vessel segmentation quantifications. There exists multiple available cost-free software for this purpose. Also, the authors should consider at analyzing myopic retina analogously to controls in this respect.
- I know that the authors show in Figure 2 a co-staining of blood vessels with astrocytes. It is certainly interesting to see how the orientation of GFAP fibers remains so variable and does not always follow along vessel guides. Could the authors quantify this and perhaps match it to which quarter of the retina was the picture taken. Do the fibers align in a specific direction with respect to the optic nerve head?
- Line 101: Please quantify when expressing “decreasing amounts of astrocytes” here.
- Line 103: Please quantify when expressing “variable widths” here. Are these widths also more associated to veins? Arteries?
- Line 105: It is necessary to introduce the concept of string vessel here and perhaps which connection has to pathology. I do think though that this is introduced too early as I only see it in Figure 3A for the myope condition. This should be definitely pointed at with in the figure later? Or use earlier in Figure 2 a panel illustrating what it is.
- Figure 2: Use the nice colored-labels to indicate which vascular plexus as in Figure 1. Also, maybe indicate in figure title that this is a control marmoset?
- Line 119-124: This study has a histopathological nature that falls too descriptive sometime. With the current availability of semi-automatic detection tools, authors should develop a method to describe with more precision the degree of “stellate” or variability in this. It is too descriptive so far.
- Figure 6C: The authors utilize this information to draw several conclusions regarding the thinning of the RGL. The quality of the images is very poor and I would have preferred to see more detailed layers in an OCT. To strengthen the conclusions, please verify the thickness using histology (vascular detection should enable this for most cases).
- Line 243-244: I guess what they refer to, but please specify the neuroretina vascular elements
- I think that in the branching analysis one should consider artery vs. vein in all cases. This is of relevance not only based on vascular biology principles but also to support some of the discussed topics on hypoxia/ischemia associated. Could the authors complement branching measurements with vascular coverage as well? This is relevant to understand blood flow implications.
- Line 308: Citation style needs to be corrected.
- Line 336: It is certainly a funny pun, but I would suggest the authors call the sub-chapter simply “conclusions”.
- The method referred to microscopy does not specify clearly the randomization used in this, nor if the analysis was done blind by a second investigator. This could imply important bias.
- Figure 7C-D: It is indicated that these boxes are shown in Figure 7B but I feel that the size does not really correlate with the represented boxes. Please correct that and if possible, include a label in the same figure to indicate that they are related to the boxes 1 or 2 in the previous panel.
- Lines 460-461: I disagree with the analysis followed using a ratio of GFAP density to Sox9. This is something complex to quantify but I don’t think that total GFAP density qualifies as a parameter to determine 1 vs. multiple astrocytes. To my understanding, Sox9 itself is a quite reliable marker in terms of “number of cells”, and GFAP signal should address “astrocyte body”, “processes” or other indicatives about the astrocyte biology. Could you suggest an alternative to this ratio for depicting the very (clear to me) result in the images?
- Line 489: It is nice to say why you say thanks to your collaborators.
Author Response
We thank reviewer #3 for their comments.
Lin and co-authors investigated vasculature architecture and astrocyte behavior in a non- human primate model for myopia using majority histological approaches. In their study, the authors found that the number of GFAP-positive cells are decreased in marmosets with myopia, and that this might be directly associated to a thinning of the RGL. They also found that vasculature branching points were reduced during myopia.
I personally find the study interesting and well-structured, and figures are organized and appealing. Understanding the neurovascular unit in disease is of relevance, and in the context of myopia still remains an understudied topic, especially using experimental primate models. The paper is well written and easy to follow with the sequence of results as displayed.
While the authors fall into a bit old-fashioned and fundamental approaches based majorly in histology (with the exception of Figure 6), I think that the data are of interest. However, most conclusions on astrocytic behavior were based on GFAP expression and this should be something to take cautiously in consideration.
Scientists are still struggling to characterize in-depth the nature and biology of astrocytes in the retina. While I agree with the conclusions of the authors based on the data shown, GFAP has been widely also associated to gliosis in Müller cells, another glial partner present in much higher numbers than astrocytes in the retina. Throughout the manuscript (results and interpretation), this fact was largely neglected or omitted. In Figure 6B, for instance, despite the much lower resolution of GFAP, it is very clear how GFAP staining is not limited to the defined astrocytic zones but also following a structure characteristic of Müller glia. Among their multiple functions, Müller glia act as scaffold in the retina, and are especially resilient cells to mechanic stress. A study relying only on GFAP associated to astrocytes and neglecting Müller glia is, from my perspective, an important bias that should be minimally (at least) addressed.
Thank you for pointing out the importance of Müller cells in our work. As the reviewer points out, the GFAP staining in the RNFL was seen both in astrocytic zones and in some areas corresponding to Muller glia end-feet, especially in myopes. While performing our 3D reconstruction of myopic retinal flatmounts, we saw that the GFAP staining was not uniform in areas corresponding to Muller cells. However, we cannot make a definitive statement on Muller cells as we did not counterstain with Muller cell markers to clearly differentiate astrocytes vs Muller cell GFAP staining. We consider your comment valuable and have addressed it in the manuscript by incorporating literature pertaining to Müller cells in the introduction and discussion, and mentioning in the results that the GFAP immunopositive staining was not limited to astrocyte changes.
Line 54-58: [Müller cells, similar to astrocytes, respond to elevated intraocular pressure[17] and retinal injury[18] with GFAP upregulation that appears greater in areas closer to ganglion cells where astrocytes and Müller cell endfeet are located. Reactive Müller cells can affect neuronal activity due to their roles in synaptic and extracellular space regulation[19]]
Line 293-298: [The increased GFAP immunopositive staining appears to affect both astrocytes and Müller glia. While performing our 3D reconstruction of myopic retinal flatmounts, the GFAP staining did not appear uniform in areas corresponding to Muller cells. We cannot make a definitive statement on Muller cells as we did not counterstain with Muller cell markers in order to clearly distinguish astrocytes versus Muller cell GFAP staining.]
Line 614-619: [Of particular importance is the role that ganglion cells, astrocytes and Müller cells play in mechanosensitivity[64], which is a feature of myopic stretch. Ganglion cell axonal damage is secondary to astrocyte, Müller, and microglia activation. They respond to mechanical stress, injury and degeneration with structural and functional changes[65], that can be detrimental or beneficial to axon survival and regeneration[66].]
I have several specific comments aiming at improving the quality of the manuscript:
- Table 1 should display also the sex of the subjects. Sex in Medicine has been a largely ignored variable for centuries and it is in our hands to start amending the lack of knowledge. Consequently, authors should look at possible sex differences in their study. 

Table 1 is now Table 2. We have added sex of the marmosets to table 2.
- Table 1: Given that all ID tags are referenced in the table, the authors could omit part of the information (e.g. age, RX) when using the ID tag in the figure legends. 

Table 1 is now Table 2. We have removed the repeated information regarding ID tags in the figure legends.
- I suggest that the authors indicate in results (when first mentioned Table 1) the threshold to consider myopia in the subjects (rx>XX) 

We have clarified that untreated marmosets on average emmetropize to low degrees of myopia (-1.50 D).
Line 724-725: [Untreated marmosets on average emmetropize to low degrees of myopia (-1.50 D)[31].]
- Figure 1B: The images are appealing but this should be complemented with semi- automatic vessel segmentation quantifications. There exists multiple available cost- free software for this purpose. Also, the authors should consider at analyzing myopic retina analogously to controls in this respect.

We have performed vessel quantification and found no differences between control and myopic marmoset retinas. The manuscript has been modified accordingly to indicate this, and the information may be found in Table 1.
Line 187-195: [The retinal vasculature coverage, as measured by IB4 staining, was quantified and no difference were found between control and myopic marmoset retinas. The percent area covered by capillaries in control and myopic animals decreased from the optic nerve head to the periphery. None of the regions differed between control and myopic marmosets (Table 1).]
Table 1: Percent Area covered by capillaries in controls and myopic marmosets (average ± SD).
|
|
Control |
Myope |
P-value |
|
Superior Periphery |
14.3 ± 3.7 |
14.0 ± 3.7 |
0.89 |
|
Superior Peripapillary |
24.3 ± 15.2 |
18.5 ± 7.0 |
0.42 |
|
Inferior Periphery |
15.2 ± 4.9 |
13.7 ± 5.0 |
0.63 |
|
Inferior Peripapillary |
19.3 ± 7.0 |
15.4 ± 5.5 |
0.35 |
|
Temporal periphery |
14.8 ± 5.3 |
13.4 ± 5.5 |
0.69 |
|
Temporal Peripapillary |
17.8 ± 5.4 |
17.4 ± 4.4 |
0.90 |
|
Nasal Periphery |
14.7 ± 3.9 |
11.6 ± 4.7 |
0.26 |
|
Nasal Peripapillary |
23.1 ± 5.0 |
17.3 ± 6.8 |
0.06 |
Line 828-834: [The percent retinal area covered by blood vessels was quantified using Fiji. The image was split into its color channels to identify the green channel corresponding to IB4 (isolectin). The green channel image was made “binary”, (Process, Binary, Make Binary), then converted to mask (Process, Binary, Convert to Mask). The resultant image is white with black particles, and the black particles were summarized (Analyze, Analyze Particles, Ok). Fiji then gave objectively the percent area covered by blood vessels.]
- I know that the authors show in Figure 2 a co-staining of blood vessels with astrocytes. It is certainly interesting to see how the orientation of GFAP fibers remains so variable and does not always follow along vessel guides. Could the authors quantify this and perhaps match it to which quarter of the retina was the picture taken. Do the fibers align in a specific direction with respect to the optic nerve head? 

The GFAP fibers are oriented elongated and radially when they leave the optic nerve head. As the GFAP fibers approach the periphery, they become less elongated and more stellate with less organization of the fibers. This description is qualitative, and found in both control and myopic retinas. Unfortunately, it is outside of our expertise to quantify GFAP orientation, and this will be removed if the reviewer considers it appropriate.
- Line 101: Please quantify when expressing “decreasing amounts of astrocytes” here. 

We are currently working to develop a reliable and repeatable method to quantify astrocyte morphology. However, at this time, it is outside the scope of this manuscript and we have removed all morphology descriptions.
- Line 103: Please quantify when expressing “variable widths” here. Are these widths also more associated to veins? Arteries? 

We have quantified vessel width and found that the average vein vessel width in focal regions 1-3 was 18.06±2.9 microns, while the average artery width in the same regions was 15.74±3.1 microns (P>0.05). The average vein vessel width in focal regions 4-6 was 12.25±3.4 microns, while the average artery width in the same regions was 9.95±2.2 microns (P>0.05).
Line 139-143: [Specifically, the average vein vessel width in focal regions 1-3 was 18.06±2.9 microns, while the average artery width in the same regions was 15.74±3.1 microns (P>0.05). The average vein vessel width in focal regions 4-6 was 12.25±3.4 microns, while the average artery width in the same regions was 9.95±2.2 microns (P>0.05).]
- Line 105: It is necessary to introduce the concept of string vessel here and perhaps which connection has to pathology. I do think though that this is introduced too early as I only see it in Figure 3A for the myope condition. This should be definitely pointed at within the figure later? Or use earlier in Figure 2 a panel illustrating what it is. 

We have added a small description of string vessels and how they relate to pathology. We have also indicated earlier in figure 3A, using white arrows, examples of string vessels in both the control and myopic vasculature.
Line 146-151: [String vessels are thin, non-functional connective tissue strands conserved across species that are remnants of capillaries. Vascular conditions like diabetes and ischemia exhibit string vessels across various capillary beds of the body[30] and their numbers increase with normal aging, but increase drastically in the presence and progression of vascular dysfunction. String vessels were identified in the superficial capillary plexus of marmoset retinas, and shown in figure 3A as white arrows.]
- Figure 2: Use the nice colored-labels to indicate which vascular plexus as in Figure 1. Also, maybe indicate in figure title that this is a control marmoset? 

Figure 2 was taken from the superficial plexus and this has now been clarified using a colored-label. Our quantification was performed for both RPC and superficial plexus. We have also clarified that Figure 2 is taken from a control marmoset.
- Line 119-124: This study has a histopathological nature that falls too descriptive sometime. With the current availability of semi-automatic detection tools, authors should develop a method to describe with more precision the degree of “stellate” or variability in this. It is too descriptive so far. 

We are currently working to develop a reliable and repeatable method to quantify astrocyte morphology. However, at this time, it is outside the scope of this manuscript and we will remove all morphology descriptions.
- Figure 6C: The authors utilize this information to draw several conclusions regarding the thinning of the RGL. The quality of the images is very poor and I would have preferred to see more detailed layers in an OCT. To strengthen the conclusions, please
verify the thickness using histology (vascular detection should enable this for most cases).
We have included a new figure (figure 6) that includes marmoset retinal histology and OCT along with a sample of automated segmentation. Below is a table summarizing the average thicknesses quantified using histology and OCT for your reference.
|
Layer |
Histology (AVG±SD microns) |
OCT (AVG±SD microns) |
|
RNFL + GCL |
11.7±0.6 |
12.3±0.6 |
|
IPL |
29.0±1.0 |
28.0±2.6 |
|
INL |
43.0±1.7 |
36.7±4.2 |
|
OPL |
11.7±1.5 |
11.3±2.1 |
|
ONL |
51.0±2.6 |
55.7±1.2 |
|
I/OS PR |
17.0±2.6 |
16.3±1.5 |
|
RPE |
35.0±1.0 |
34.7±5.5 |
|
Choroid |
67.0±3.6 |
70.3±1.5 |
- Line 243-244: I guess what they refer to, but please specify the neuroretina vascular elements 

We have changed the terminology to “neuroretinal function” instead of “neural function”.
Line 682-686: [However, since astrocytes and retinal capillaries are crucial to maintain RGC axonal viability[26,79], the reduced astrocyte numbers observed in myopic eyes, along with the increased GFAP immunopositive staining, reduced capillary branching and increased number of string vessels might be affecting neuroretinal function.]
- I think that in the branching analysis one should consider artery vs. vein in all cases. This is of relevance not only based on vascular biology principles but also to support some of the discussed topics on hypoxia/ischemia associated. Could the authors complement branching measurements with vascular coverage as well? This is relevant to understand blood flow implications. 

Due to the magnification at which the branching analysis was performed, we were unable to distinguish arteries from veins. We have clarified this in the results and discussion.
Line 184-185: [Due to the magnification at which the branching analysis was performed, we were unable to distinguish arteries from veins.]
Line 526-529: [In our study, due to the magnification at which the branching analysis was performed, we are unable to distinguish arteries from veins. Thus, our discussion is limited regarding whether arteries or veins branched more in the myopic retinas.]
- Line 308: Citation style needs to be corrected. 

We have corrected the style of this reference.
- Line 336: It is certainly a funny pun, but I would suggest the authors call the sub- chapter simply “conclusions”. 

We re-named the sub-chapter “conclusions”.
Line 666: [Conclusions]
- The method referred to microscopy does not specify clearly the randomization used in this, nor if the analysis was done blind by a second investigator. This could imply important bias. 

The analysis was performed in a randomized manner by one blinded investigator. We have added this important information to our manuscript.
Line 772-773: [The images were gathered and the analyses were performed in a randomized manner by one blind investigator.]
- Figure 7C-D: It is indicated that these boxes are shown in Figure 7B but I feel that the size does not really correlate with the represented boxes. Please correct that and if possible, include a label in the same figure to indicate that they are related to the boxes 1 or 2 in the previous panel. 

We have corrected the size of the boxes in Figure 7, and have also included a label to indicate that they are related to the boxes 1 or 2 in the previous panel.
- Lines 460-461: I disagree with the analysis followed using a ratio of GFAP density to Sox9. This is something complex to quantify but I don’t think that total GFAP density qualifies as a parameter to determine 1 vs. multiple astrocytes. To my understanding, Sox9 itself is a quite reliable marker in terms of “number of cells”, and GFAP signal should address “astrocyte body”, “processes” or other indicatives about the astrocyte biology. Could you suggest an alternative to this ratio for depicting the very (clear to me) result in the images? 

The alternative to this ratio is to present astrocyte cell counts using Sox9 and astrocyte coverage using GFAP to study activation. We have modified our methods, results, and discussion to indicate this.
Line 280-284: [Despite the reduced astrocytes cell counts, the spatial coverage of astrocyte processes quantified as percent GFAP positive (GFAP+) immunostaining was significantly greater in the peripapillary but not the peripheral retina of myopes compared to controls (Figure 4C, Peripapillary p<0.05; Periphery P>=0.70).]
Line 290-294: [The percent GFAP+ immunostaining was greater in the RPC and superficial vascular plexi of myopic foveas and parafoveas (RPC Figure 5D, Fovea P<0.05, Parafovea P<0.05; superficial Figure 5F, Fovea P<0.05, Parafovea P<0.05). The increased GFAP immunopositive staining appears to affect both astrocytes and Müller glia.]
Line 611-614: [In this study, a significant increase in GFAP immunopositive staining was observed in the myopic parafovea, suggesting a mild astrocyte and Müller cell activation and a potential compromised glial support to the ganglion cells of myopic eyes.]
Line 835-843: [The number of astrocyte nuclei was counted in every image frame using Fiji cell counter function and converted to astrocytes/mm2. The image was split into its color channels to identify the red channel corresponding to GFAP. The red channel image was made “binary”, (Process, Binary, Make Binary), then converted to mask (Process, Binary, Convert to Mask). The resultant image is white with black particles, and the black particles were summarized (analyze, Analyze Particles, Ok). Fiji then gave objectively the percent area of GFAP coverage, which we interpreted as GFAP immunopositive staining and which we used to quantify astrocyte spatial coverage.]
- Line 489: It is nice to say why you say thanks to your collaborators. 

We have given thanks for the guidance each of our collaborators have given us through the course of this manuscript.
Line 886-889: [Stefanie Wohl for her advice on immunohistochemical techniques, Harrison Feng for his role in initial conceptualization of the project, Rita Nieu, Amy Pope, and Victor Lin for their help to treat myopic marmosets, and Ana Nour, Xiomara Santiago, and Mirella Camargo for her attention and care to the marmosets included in our study.]
Round 2
Reviewer 3 Report
Thank you to the authors for their kind response and for incorporating many of my suggestions to their manuscript. I believe that these changes have overall improved the quality of the work.
Since the authors agreed that GFAP is an inaccurate marker for astrocytes, I would prefer if the title would be modified accordingly towards something more directed to evaluation of GFAP-positive cells instead.
I am still concerned about some implications derived from the only-qualitative evaluation of IHC data (GFAP). The authors mention that they are working on another project to quantify astrocyte morphology but this this is out of the scope of this paper. I would keep data and comments regarding astrocyte morphology to the minimum, and indicate clearly the intention of evaluating these features quantitatively in future works. I believe that establishing a connection with the phenotype of astrocytes (more stellate or more polygonal?) with disease could be of use not only in this context (myopia) but also in other retina diseases. It is unfortunate that this cannot be included in this paper.
I like the new Figure 6, but I think that the authors should include letters to identify the panels more easily though and refer to them accordingly in the figure legend.
In the new Figure 7, it would be preferable that panel A has the current length OCT, but also a more close-up to the perifoveal and foveal region in the OCT image as in the previous version (Figure 6 when first-submitted). Perhaps this can be incorporated as an insert in the panel. As for panel B, did the authors by chance had done the IHC with DAPI? The orientation of the retina should be indicated to understand better the staining.
Given that the marmosets were screened with SD-OCT, do the authors have additional functional data of these animals? What about the other retinal layer thickness quantification (only Fiber layer and GCL are shown)? Even if not differences, I would add this to the supplemental of this manuscript.
Author Response
Thank you to the authors for their kind response and for incorporating many of my suggestions to their manuscript. I believe that these changes have overall improved the quality of the work.
We thank you very much for your suggestions and efforts to improve our work, they were very valuable. We agree that the manuscript has improved with your additions.
Since the authors agreed that GFAP is an inaccurate marker for astrocytes, I would prefer if the title would be modified accordingly towards something more directed to evaluation of GFAP-positive cells instead.
We have modified the title from “Myopia alters the structural organization of the retinal astrocyte template, associated vasculature and ganglion layer thickness” to “Myopia alters the structural organization of the retinal vasculature, GFAP-positive glia and ganglion cell layer thickness.”
Line 2-4: [Myopia alters the structural organization of the retinal vasculature, GFAP-positive glia and ganglion cell layer thickness]
I am still concerned about some implications derived from the only-qualitative evaluation of IHC data (GFAP). The authors mention that they are working on another project to quantify astrocyte morphology but this this is out of the scope of this paper. I would keep data and comments regarding astrocyte morphology to the minimum, and indicate clearly the intention of evaluating these features quantitatively in future works. I believe that establishing a connection with the phenotype of astrocytes (more stellate or more polygonal?) with disease could be of use not only in this context (myopia) but also in other retina diseases. It is unfortunate that this cannot be included in this paper.
We have kept terminology related to astrocyte morphology to a minimum and added a sentence indicating the intention to evaluate quantitatively the morphology of astrocytes in future work. We agree that finding the connection between astrocyte phenotype and disease would be useful and hope to provide this information in time.
Line 729-731: [Future studies intend to evaluate quantitatively the morphology of astrocytes and how astrocyte phenotype may relate to retinal disease.]
I like the new Figure 6, but I think that the authors should include letters to identify the panels more easily though and refer to them accordingly in the figure legend.
We have included letters to identify the panels more easily, and adjusted them accordingly in the figure legend.
Lines 412-418: [Figure 6: (A) A control marmoset retinal histology slice (ID tag H16 Right), compared to (B) a control marmoset SD-OCT scan (ID tag C16 Right). (C) An example of automated OCT segmentation from Iowa Reference Algorithms v3.6, performed on the same OCT scan from (B). (D) A side-by-side of histology and OCT scan, at higher magnification, of (A) and (B). Yellow (from histology) and green (from OCT) squares indicate areas that have been magnified to allow a direct comparison of the retinal layers.]
In the new Figure 7, it would be preferable that panel A has the current length OCT, but also a more close-up to the perifoveal and foveal region in the OCT image as in the previous version (Figure 6 when first-submitted). Perhaps this can be incorporated as an insert in the panel. As for panel B, did the authors by chance had done the IHC with DAPI? The orientation of the retina should be indicated to understand better the staining.
Thank you for your comment. We have included the less magnified OCT image from the initial submission, as well as the close-up of the perifoveal/foveal regions from the second submission for easier viewing. Unfortunately, we did not stain with DAPI since we were only able to use 3 IHC markers per marmoset retina (GFAP, Sox9 and IB4).
Line 484-514: [Figure 7. OCTs from the control and myopic marmoset. The graphs show differences in RNFL layer thickness, and the GFAP RFI of different retinal regions to be higher in the myopic retina. (A) Average ganglion cell layer thicknesses in micrometers (mm) in the quadrants around the fovea of control (top left panel) and myope (bottom left panel) marmosets, as measured with SD-OCT. Representative image of en face optical coherence tomography around the fovea of a control marmoset (top left: ID tag C16 Right) and myopic (bottom left: ID tag O17 Left). Representative cross-sectional scan images of the fovea of the same marmosets can be seen for the control marmoset (top right panel, top) and myopic marmoset (bottom right panel, top), with close-ups of the cross-sectional scan images to be found in the pink box for the control (top right panel, bottom) and the blue box for the myope (bottom right panel, bottom). (B) Representative GFAP RFI images of the superficial retina in a control (top: H16 Right; bottom: P17 Right), showing increased GFAP RFI in the myopic retina compared to the control. (C) Analysis of the ganglion cell layer thickness showed that the myopic GCL was no different to the control GCL thickness in the parafoveal retina (P=0.13). However, there is a significant decrease in the myopic RNFL thickness, compared to that of the control RNFL thickness (P=0.04). (D) GFAP RFI was significantly increased in the myopic peripheral superior, inferior, and nasal retinas (Peripapillary P=0.12, Periphery P<0.05) (E) GFAP RFI was significantly increased in the myopic peripheral temporal retina (Peripapillary P=0.06, Periphery P<0.05). (F) GFAP RFI was significantly increased in the myopic parafoveal and foveal retinas (Parafovea P<0.001, Fovea P<0.05).]
Given that the marmosets were screened with SD-OCT, do the authors have additional functional data of these animals? What about the other retinal layer thickness quantification (only Fiber layer and GCL are shown)? Even if not differences, I would add this to the supplemental of this manuscript.
We have electroretinogram data (ERG) data on our marmosets. We are analyzing it and hope to submit that work for publication soon. The preliminary analysis shows evidence of very early alterations in the photonegative response (PhNR) in marmosets with myopia compared to control animals. These PhNR changes may reflect early changes in ganglion or glial cell physiology during myopia development. We are also working on analyzing full retinal OCTs over the course of 6 months during myopia development in marmosets. Our preliminary analysis points towards a normal retinal thickening as untreated control eyes grow that differs from the thinning of the inner neuroretina and GCIPL observed in myopic eyes as they develop myopia. For this manuscript, we focused on the RNFL and the GCL due to their known relationship to astrocytes.